# Transcriptomic Analysis Reveals Candidate Genes in Response to Sorghum Mosaic Virus and Salicylic Acid in Sugarcane

**DOI:** 10.3390/plants13020234

**Published:** 2024-01-14

**Authors:** Genhua Zhou, Rubab Shabbir, Zihao Sun, Yating Chang, Xinli Liu, Pinghua Chen

**Affiliations:** 1Key Laboratory of Sugarcane Biology and Genetic Breeding, Ministry of Agriculture, National Engineering Research Center for Sugarcane, Fujian Agriculture and Forestry University, Fuzhou 350002, China; mszhougenhua@163.com (G.Z.); rubabshabbir28@gmail.com (R.S.); jeremy_szh@126.com (Z.S.); cyating0302@163.com (Y.C.); 18350065738@163.com (X.L.); 2Key Laboratory of Ministry of Education for Genetics, Breeding and Multiple Utilization of Crops, College of Agriculture, Fujian Agriculture and Forestry University, Fuzhou 350002, China

**Keywords:** sugarcane, sorghum mosaic virus, salicylic acid, transcriptomics

## Abstract

Sorghum mosaic virus (SrMV) is one of the most prevalent viruses deteriorating sugarcane production. Salicylic acid (SA) plays an essential role in the defense mechanism of plants and its exogenous application has been observed to induce the resistance against biotic and abiotic stressors. In this study, we set out to investigate the mechanism by which sorghum mosaic virus (SrMV) infected sugarcane responds to SA treatment in two sugarcane cultivars, i.e., ROC22 and Xuezhe. Notably, significantly low viral populations were observed at different time points (except for 28 d in ROC22) in response to post-SA application in both cultivars as compared to control based on qPCR data. Furthermore, the lowest number of population size in Xuezhe (20 copies/µL) and ROC22 (95 copies/µL) was observed in response to 1 mM exogenous SA application. A total of 2999 DEGs were identified, of which 731 and 2268 DEGs were up- and down-regulated, respectively. Moreover, a total of 806 DEGs were annotated to GO enrichment categories: 348 biological processes, 280 molecular functions, and 178 cellular components. GO functional categorization revealed that DEGs were mainly enriched in metabolic processes, extracellular regions, and glucosyltransferase activity, while KEGG annotation revealed that DEGs were mainly concentrated in phenylpropanoid biosynthesis and plant-pathogen interaction suggesting the involvement of these pathways in SA-induced disease resistance of sugarcane in response to SrMV infection. The RNA-seq dataset and qRT-PCR assay showed that the transcript levels of *PR1a*, *PR1b*, *PR1c*, *NPR1a*, *NPR1b*, *PAL*, *ICS,* and *ABA* were significantly up-regulated in response to SA treatment under SrMV infection, indicating their positive involvement in stress endorsement. Overall, this research characterized sugarcane transcriptome during SrMV infection and shed light on further interaction of plant-pathogen under exogenous application of SA treatment.

## 1. Introduction

Environmental stressors brought on by extreme climate change are becoming more prevalent worldwide and have also affected the geographic distribution and incidence of plant pathogens and related diseases, which by the end of this century could push about one-third of the world’s food crop productivity outside of the safe climatic range [1]. Various biotic stresses are major factors that can impede the whole growth and development period of sugarcane, resulting in 10–15% yield losses worldwide [1]. Plants recognize the first layer of conserved pathogen-associated molecular patterns (PAMPs) through the pattern-recognition receptors (PRRS) to activate pattern-triggered immunity (PTI) in response to various stresses [2,3]. In order to activate effector-triggered immunity (ETI), plants selectively detect effectors using polymorphic nucleotide-binding domain and leucine-rich repeat (NB-LRR) proteins expressed by the majority of resistance genes [2,3,4].

Numerous transcriptomic approaches have been undertaken in sugarcane infected with various biotic and abiotic stresses. For pathogen infection, several studies have been undertaken on sugarcane infected by sugarcane yellow leaf virus (SCYLV) [5], sugarcane streak mosaic virus (SCSMV) [6], *Sporisorium scitamineum* [7], *Acidovorax avenae* subsp. *avenae* (Aaa) [8] and *Xanthomonas albilineans* (*Xa*) [9]. Comparative transcriptome analysis of resistant and susceptible sugarcane cultivars (LCP85-384 and ROC20) evidenced the upregulated expression of numerous *WRKYs* genes especially for *WRKY33* alleles against *Xa* infection [9]. The transcriptional regulatory co-expression network revealed 16 and 25 hub genes in *Saccharum spontaneum* (SES208) that were enriched for possible involvement in MAPK signaling, plant-pathogen interactions, and hormonal signaling against resistance to leaf scald [10].

Several studies have revealed that the exogenous application of SA regulates ROS homeostasis, GR (glutathione reductase), and APX (ascorbic acid-peroxidase) activity and ultimately increases the resistance against biotic and abiotic stressors [11]. The SA application also induces the PR-proteins against *V. dahlia* in *S. melongena* [12] which in turn regulate resistance against *P. palmivora* by increasing the activity of PAL (phenylalanine ammonialyase), POD (peroxidase), and CAT (catalase) in rubber trees [13]. Another research revealed that SA application could increase the APX and POD activity to enhance resistance against *tomato yellow leaf curl virus* [14]. An increasing number of studies revealed that SA pathway PR1 proteins play a pivotal role in the response of plants to pathogen infection. For instance, ectopic expression of *Mangifera indica MiPR1A* in transgenic *Arabidopsis* attributed to enhanced disease resistance to *Colletotrichum gloeosporioides* pathogen [15]. Overexpression and virus-induced gene silencing of *Triticum aestivum TaPR1-7* and *Solanum lycopersicum SlPR1* exhibited decreased pathogenesis caused by *Puccinia striiformis* f. sp. *tritici* and *Ralstonia solanacearum*, respectively [16,17]. The SA has the advantages of high efficiency, non-toxicity, broad-spectrum, environmental safety, simple use, and low cost, and has become a research hotspot in the field of resistance induction to plant diseases [18,19].

Single and/or mixed viral disease pathogens such as sugarcane mosaic virus (SCMV), sorghum mosaic virus (SrMV), sugarcane streak mosaic virus (SCSMV), and maize yellow mosaicvirus (MaYMV) causing significant loss to the global sugar industry. These viruses have a wide range of hosts. i.e., maize, sorghum, and other grasses in addition to sugarcane. SrMV (genus: Potyvirus, family: Potyviridae) is a causal agent of common mosaic in sugarcane. Additionally, SrMV symptoms include chlorotic streaks on leaves, stunted growth, necrosis, and death of the plant [20].

However, only gene profiles of sugarcane responses to a single stressor alone have been considered. To our knowledge, no report relates to an integrated transcriptome analysis of SrMV infection and SA in sugarcane. Therefore, we investigated the molecular responses in sugarcane cultivar Xuezhe under SrMV infection and SA application based on the RNA-seq platform by Illumina NGS technology. Genes and pathways involved in response to the coupling effect of SrMV infection and SA application as part of a defense strategy against this pathogen were identified. These potential genes serve as helpful genetic resources for sugarcane breeding as well as provide insights into the signaling cascades activated by SrMV infection under SA enrichment.

## 2. Results

### 2.1. SA Application Inhibited SrMV from Infecting Sugarcane Plants

Based on qPCR data, significantly low viral populations were observed at different time points (except for 28 d in ROC22) in response to post-SA application in both cultivars as compared to control (no SA application). The lowest number of population size in Xuezhe (20 copies/µL) and ROC22 (95 copies/µL) was observed in response to 1 mM exogenous SA application as compared to other SA treatments and controls. In contrast to the aforementioned results, the highest number of population size was identified in ROC22 (1209 copies/µL) in response to a higher dose of SA (4 mM SA) at 28 d post-SrMV infection. No variance in viral population size was observed at 14 d post-SrMV infection in response to different SA treatments in ROC22. Overall, the results indicated the positive involvement of exogenous SA application in decreased population size of SrMV in both sugarcane cultivars (Figure 1).

### 2.2. Transcriptome Sequencing and Assembly

To investigate the genes and pathways involved in sugarcane response to SrMV infection and SA treatment, a total of 12 cDNA libraries were screened (Table 1). The number of raw reads and clean data after quality control ranged from 40,056,842 to 47,349,582 and 37,546,314 to 45,324,754 for the Xuezhe cultivar, respectively. The GC content, Q20, and Q30 of 12 libraries was more than 56%, 97%, and 93% respectively. Moreover, the mapping rate ranged between 78.52% and 89.37% (Table 1).

### 2.3. Identification, Functional Annotation of DEGs, Gene Ontology (GO), and Kyoto Encyclopedia of Genes and Genomes (KEGG) Analysis

A total of 2999 DEGs were identified in treatment comparison with control of which 731 DEGs were upregulated while 2268 DEGs were downregulated. Go analysis revealed that the top 10 upregulated and downregulated transcripts were assigned to one of the three main GO categories: cellular component (CC), biological process (BP), and molecular function (MF). A total of 806 DEGs were annotated, including 348 BP, 280 MF, and 178 CC. The DEGs were further distributed to 30 main GO groups. Notably, 204 DEGs (51 for each ontology) were attributed to the cellular glucan metabolic process, glucan metabolic process, cellular polysaccharide metabolic process, and polysaccharide metabolic process of the BP category. Similarly, 39 and 36 DEGs were attributed to the extracellular region and cell periphery of the CC category. Additionally, glucosyltransferase activity, oxidoreductase, antioxidant activity, and peroxidase activity had 52, 43, 43, and 42 DEGs, respectively, in the MF category (Figure 2).

To further study the biological pathways triggered by SrMV infection and SA application, DEGs were annotated by KEGG enrichment analysis. Overall, the most significant 20 KEGG pathways were selected (Figure 3). The results showed that DEGs are mainly concentrated in phenylpropanoid biosynthesis, plant-pathogen interaction, benzoxazinoid biosynthesis, fatty acid elongation, thiamine metabolism, alpha-linolenic acid metabolism, and zeatin biosynthesis. The enrichment of DEGs was most significant in phenylpropanoid biosynthesis and plant-pathogen interaction, suggesting that that pathway may be involved in SA-induced disease resistance of sugarcane (Figure 3).

### 2.4. Transcript Profiling of Candidate Genes

Based on transcript expression of genes in RNA-seq data, 8 DEGs were selected for validation by qRT-PCR assay (Figure 4 and Figure 5). The characterization of genes responding to SrMV infection and SA was determined by qRT-PCR assay. The transcript levels of 3 *PR1s* (*PR1a*, *PR1b*, and *PR1c*), 2 *NPR1s* (*NPR1a* and *NPR1b*), *PAL*, *ICS,* and *ABA* were significantly up-regulated after the SA treatment in response to SrMV.

## 3. Discussion

Climate change is predicted to have significant effects on sugarcane production around the globe, particularly in developing countries because of relatively low adaptive capacity, high vulnerability to natural hazards, limited gene pool, and poor forecasting systems as well as preventive/mitigating measures. Sugarcane production may have been negatively affected and will continue to be considerably affected by increases in the frequency and intensity of extreme environmental conditions such as biotic and abiotic stresses [1,2,3,4,5,6,7,8,9]. Over the last decade, the advent of next-generation sequencing (NGS) such as transcriptomics, metabolomics, genomics, and proteomics have emerged as promising tools for exploring genetic and molecular basis to identify networks of regulatory candidate genes and biosynthetic cascades under the ongoing climate change scenario in sugarcane [6,7,8,9]. The combination of sugarcane omics technologies through identification and modifications in transcripts, metabolites, DNA, and proteins enables a deeper understanding of the genetic mechanisms underlying the complex architecture of many phenotypic/genotypic traits of sugarcane relevance [7,8,9]. This study also employed a transcriptomic approach to identify key and candidate genes responding to SrMV infection coupled with SA application.

### 3.1. SA Pathway and Regulatory Genes

SA plays an important role in all aspects of plant immunity and can effectively promote plant growth and development [2]. The effects of SA on plant growth and development are dependent on the concentration, plant growth conditions, and different developmental stages. Usually, high SA concentration negatively regulates plant growth while the application of optimal SA concentration shows beneficial effects. Exogenous application of SA could significantly promote plant growth and increase plant dry yield under both salt stress and no-stress conditions, and the effect was more obvious under salt stress [21]. Exogenous application of SA could significantly reduce the infection effect of the virus and effectively increase plant height and leaf number [22]. Similar findings were reported by Zagier et al. [23] to control fig mosaic virus by exogenous application of SA. Based on qPCR results, low viral populations were observed at different time points (except for 28 d in ROC22) in response to post-SA application in both cultivars as compared to the control in this study. Similarly, Suharti et al. [24] found that after SA induction, *Xanthomonas oryzae* infection was inhibited and plant biochemical resistance increased, especially tannins and phenols. However, salicylic acid did not affect the growth of rice plants which might be caused by species differences.

The SA signaling pathway is involved in dramatic transcriptional reprogramming against environmental stimuli in plants via several important genes such as *NPR1*/*NPR3* and disease process-related proteins PR1/5, etc. SA signals play a key role in triggering defense responses to stress and activating plant systemic acquired resistance (SAR). In SA signal transduction, up-regulated expression of SAR-related genes leads to increased plant disease resistance. Zhang et al. [25] used RNA-seq to study the differential expression of 32 SA signal-related genes, including *NPR1*, *TRXs*, *NPR3*, *NIMINs*, *WRKYs*, *TGAs*, *SABP2* and *MESs* genes. NPR1 is a major regulator in the SA signaling pathway, controlling multiple immune responses including SAR. In NPR1 mutants, SA-mediated PR gene expression and pathogen resistance were eliminated. In the inactivated state, *NPR1* exists in the cytoplasm as a disulfide-bonded oligomer. After induction by exogenous SA, the cytoplasmic TRX catalyzes the REDOX of *NPR1* from an oligomeric form to a monomer form. The monomer form of *NPR1* can then enter the nucleus and regulate downstream transcription factors, such as TGA and WRKY. Plant WRKY TFs are involved in regulating disease-resistance responses at multiple levels, such as directly or indirectly regulating downstream target genes, activating, or inhibiting the expression of other TFs, and regulating WRKY family members and self-genes using multiple feedforward and feedback methods. WRKYs have been shown to play a major positive or negative regulatory role in SA-dependent defense responses in plants, with more than half of *Arabidopsis* WRKY genes induced or inhibited when treated with SA [26]. For example, transcriptional regulator *AtWRKY70* signals through the SA-mediated pathway and thereby regulates a subset of genes common to senescence and plant defense [27]. SA-induced *AtWRKY18* positively regulates the expression of the PR gene and resistance to *Pseudomonas syringae* as well as *AtWRKY18* enhances the developmental regulatory defense response dependent on *NPR1* [28]. A recent study by Chu et al. [29] reported the involvement of *ScPR1* as a positive regulator of defense response against *Aaa* infection in sugarcane. Previously, it has been found that *NPR3* binds to *NPR1* and TGAs to regulate the defense response of developing flowers to pathogens [30]. Notably, SA-mediated genes (*ScNPR3*, *ScTGA4*, *ScPR1*, and *ScPR5*), participated in the response of sugarcane to infection by *Xa* strains, but sugarcane responses to two strains differing in pathogenicity through differential modulation of SA and ROS [31].

Transcription factors are ubiquitous regulatory proteins in plants, which play a regulatory role in plant growth and development, biological and abiotic stress response, hormone response, and metabolism [32]. In this study, by further analysis of DEGs, it was found that WRKY and GST transcription factor families were highly expressed after SA treatment. WRKY transcription factor is one of the ten major transcription factor families in plants. TFs are activated by different pathways of signal transduction and can directly or indirectly combine with *cis*-acting elements to modulate the transcription efficiency of target genes, which play key regulators in crop genetic improvement. Interestingly, WRKY TFs are widely involved in plant disease resistance response, plant growth, development and physiological processes, and regulatory networks affecting plant secondary metabolic pathways [33,34]. When plants suffer from stress, the antioxidant activity in the plant is enhanced, among which GST is one of the main antioxidant enzymes, which can degrade harmful substances produced in plants and reduce cell damage. In this study, the large expression of the GST transcription factor can promote the production of this enzyme and thus enhance the plant’s defense against disease. In conclusion, the large expression of WRKY and GST transcription factors may be the main response mode of SrMV resistance induced by SA in sugarcane in this experiment, which can be further studied in the future. In this study, it was also found that the ABA pathway genes were also significantly up-regulated after the treatment. ABA has been reported to play a role in abiotic and biological stress through bidirectional JA and SA crosstalk [35].

### 3.2. Metabolic Adjustments

Phenylalanine ammonia-lyase converts phenylalanine into trans-cinnamic acid to synthesize SA in a distinct enzymatic pathway. The SA signaling pathway can be triggered both by PTI and ETI. Moreover, plants have developed a multitude of multiple genes regulated defense responses such as PAMP by modulating different metabolic pathways i.e., galactose-, phenylpropanoid biosynthesis-, and amino acids-metabolisms [33]. These responses also include activation of pathogenesis-related proteins (PR), production of defense markers (reactive oxygen species), and signaling dependent on the phytohormones [30,31]. Under *Aaa* infection, increased expression of several genes involved in the phenylpropanoid biosynthesis pathway would be a defense strategy employed by sugarcane against pathogen stress [8]. In this study, we found that phenylpropanoid biosynthesis pathway genes were significantly enriched under SrMV infection coupled with SA as compared to control, suggesting that SA signaling cascade may also exist in sugarcane to positively regulate SrMV pathogenesis.

## 4. Conclusions and Prospects

This study provides the first report on a transcriptome dataset of 12 cDNA libraries in response to SrMV infection and SA enrichment in sugarcane. Most of the DEGs were annotated for cellular glucan metabolic process, glucan metabolic process, cellular polysaccharide metabolic process, and polysaccharide metabolic process. KEGG pathway analysis revealed that phenylpropanoid biosynthesis and plant-pathogen interaction pathways contributed to sugarcane response to SrMV infection coupled with SA supply. In this study, 8 up-regulated genes were selected for qRT-RCR validation: 3 *PR1*, 2 *NPR1*, 1 *PAL*, 1 *ICS,* and 1 *ABA*. The results showed that the expression profiles of several genes were consistent with the expression trend of the transcriptome. These results provide data support for further understanding of the SA regulatory network in sugarcane and the molecular mechanism of SA regulating defense and stress response at the transcriptional level and lay a foundation for revealing the systematic gene functions and protein interactions in the SA signaling pathway in sugarcane under SrMV pathogenesis. Additionally, this study also lays the foundation for the identification of new gene resources in response to SrMV infection coupled with SA application.

## 5. Materials and Methods

### 5.1. Plant Materials and Virus Detection

The experimental materials were selected from the SrMV-sensitive sugarcane variety Xuezhe and the SrMV-resistant sugarcane variety ROC22, both of which were provided by the National Sugarcane Germplasm Resource Nursery in Kunming, Yunnan Province, China. Sugarcane stems were cut into single-budded setts followed by imbibition in water for 24 h at room temperature and incubation at 32 °C till seedlings emerged and grew up to a height of 3–5 cm. The seedlings were transplanted to nutrient soil in an incubator with the following specifications: 28 °C temperature, 16 h of light/8 h of darkness, and 6000 Lx of light intensity [18]. SA solution with concentrations of 0 mM (CK), 1 mM, 2 mM, and 4 mM was prepared and applied to sugarcane seedlings soon after SrMV inoculation. The samples of Xuezhe and ROC22 seedlings inoculated with SrMV were taken at 7, 14, 21, and 28 days after inoculation. The SrMV content in sugarcane leaves at different periods was determined using the detection method established by the standard curve (attachment) constructed in the earlier stage of the laboratory (Appendix A).

### 5.2. RNA Extraction and cDNA Library Construction

The healthy seedlings of sugarcane cultivar Xuezhe at the 3–5 leaf stage were selected for infection by SrMV. The leaf cut method was employed to infect the leaves with SrMV for 28 days, and each treatment had 6 biological repeats. SA solution with a concentration of 1 mM was prepared and applied to sugarcane seedlings following SrMV inoculation. The plant leaf samples were used for transcriptome analysis. RNA samples were sent to Novogene Biological Company, Beijing, China, for transcriptome sequencing analysis. According to the NEBNext^®^ Ultra ™ RNA Library Prep Kit for Illumina^®^ (San Diego, CA, USA) manual of the kit, cDNA libraries were constructed. Finally, the sequencing was performed through the Illumina HiSeq platform.

### 5.3. Quality Control and Alignment of Sequencing Data

Trimmatic (0.39) [36] was used to remove a small number of reads with sequencing connectors, sequencing errors, and low-quality sequences from the original data. Q20, Q30, and GC contents were calculated from the clean data. Genome (https://download.cncb.ac.cn/gwh/Plants/Saccharum_officinarum_LApurple_GWHBEII00000000/GWHBEII00000000.genome.fasta.gz (accessed on 12 March 2022)) and annotation (https://download.cncb.ac.cn/gwh/Plants/Saccharum_officinarum_LApurple_GWHBEII00000000/GWHBEII00000000.gff.gz (accessed on 12 March 2022)) files LA purple from the genome website were downloaded and by using HISAT2 (2.0.5) [37] an index of the reference genome to obtain location information was calculated.

### 5.4. Screening and Functional Enrichment Analysis of Differentially Expressed Genes

StringTie (1.3.3b) [38] was used to predict new genes, and the number of fragments per thousand base length from a gene in each million reads of each gene was calculated according to the gene length. DESeq2 (1.20.0) [39] was used for differential expression analysis based on the negative binomial distribution model. The fragments per kilobase of transcript per million fragments mapped (FPKM) value for each transcript was calculated and then transformed to log2 (Fold Change) values for the generation of a heatmap with TBtools v0.6655. DEGs were screened with a *p*-value ≤ 0.05. ClusterProfiler (3.8.1) [40] software was used to conduct GO (Gene Ontology) and KEGG (Kyoto Encyclopedia of Genes and Genomes) analysis of DEGs. The criterion for significantly different expression was established at |log_2_(fold change)| > 1.0 and *p*-value < 0.005. Following transcriptome assembly, BLASTx with an E-value cutoff of 10^−5^ was used to assign functional categories in the GO and KEGG databases. Wallenius non-central hyper-geometric distribution from the GOseq R package (v.4.2.3) was used to analyze the GO enrichment of differential expression genes (DEGs). GO terms with a *p*-value of 0.005 or lower showed substantial enrichment. KOBAS 2.0 was used for the KEGG enrichment analysis of DEGs with a *p*-value < 0.05.

### 5.5. Quantitative Real-Time PCR Analysis

The RNA from leaf samples was extracted by using a megazol reagent (Invitrogen, Waltham, MA, USA) according to the manufacturer’s instructions. The extracted sample RNA was reversely transcribed into cDNA for qRT-PCR verification. A total of 8 differential genes were selected for verification by qRT-PCR. The qRT-PCR was carried out using ChamQ Universal SYBR qPCR master mix (Vazyme, Nanjing, China) on a QuantStudio^®^ Real-Time PCR system (Applied Biosystems, Waltham, MA, USA). Additionally, IDT online software (https://sg.idtdna.com/PrimerQuest/Home/Index (accessed on 12 March 2022)) was used to design gene-specific primers for qRT-PCR (Appendix A). The reaction mixture comprised 10 μL of 2× SYBR PremixEx TaqTM (Takara, San Jose, CA, USA), 2 μL of each forward and reverse primer, 1 μL of cDNA, and 6.6 μL of ddH_2_O to make a final volume of 20 μL. For each sample, three biological and three technical replicates were used. Glyceraldehyde 3-phosphate dehydrogenase (GAPDH) was used as a reference gene and 2^−ΔΔCt^ method was used to determine the gene expression.

### 5.6. Statistical Analysis

The means of different treatments were compared using the least significance difference (LSD) test at a 5% probability level (*p* ≤ 0.05) with a statistical software package Statistix 8.1. This means having different letters above the bars are significantly different at a 5% probability level.

## Figures and Tables

**Figure 1 plants-13-00234-f001:**
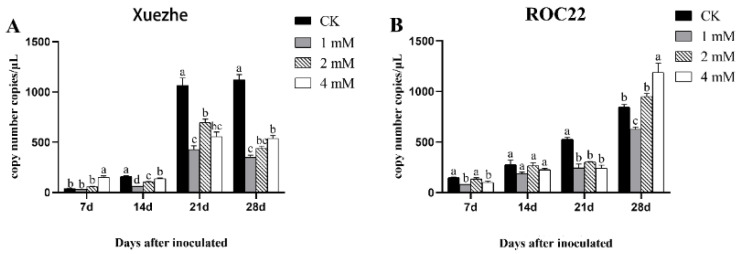
Virus populations in two sugarcane cultivars post SA application determined by qPCR assay: (**A**,**B**) represent the viral copy numbers of Xuezhe and ROC22, respectively. SA solution with concentrations of 0 mM (CK), 1 mM, 2 mM, and 4 mM was prepared and applied to sugarcane seedlings following SrMV inoculation. The letters on the bar graph indicate significant differences among different treatment means (*p* < 0.05), the same letters indicate no significant differences, and different letters indicate significant differences. Values are means ± standard errors.

**Figure 2 plants-13-00234-f002:**
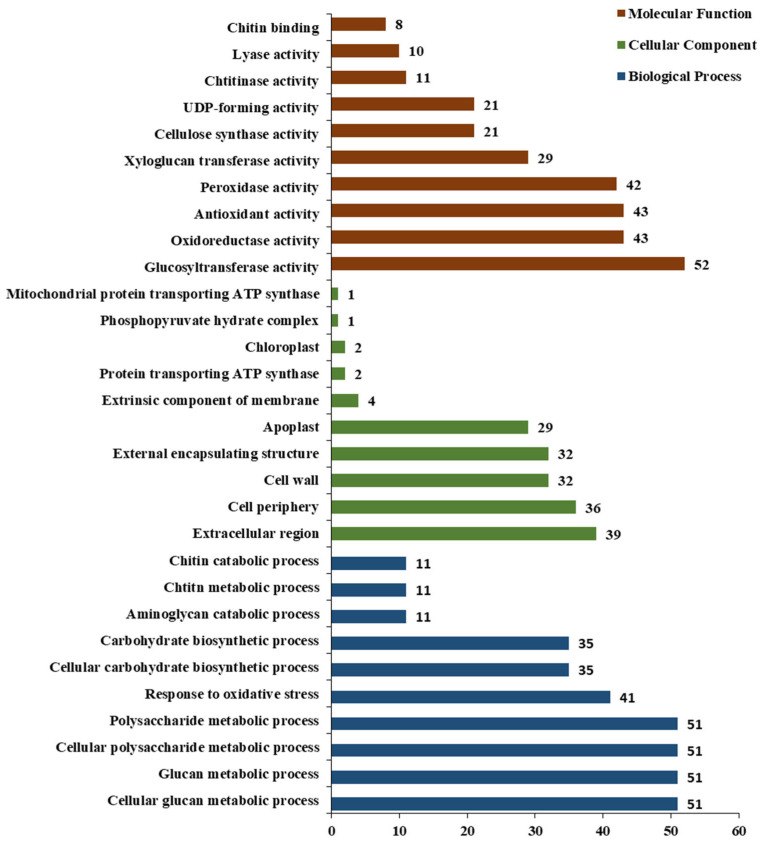
Differentially expressed genes GO enrichment analysis. Note: The abscissa is the GO term, and the ordinate is the significance level of GO term enrichment, which is represented by −log10(padj), and different colors represent different functional categories.

**Figure 3 plants-13-00234-f003:**
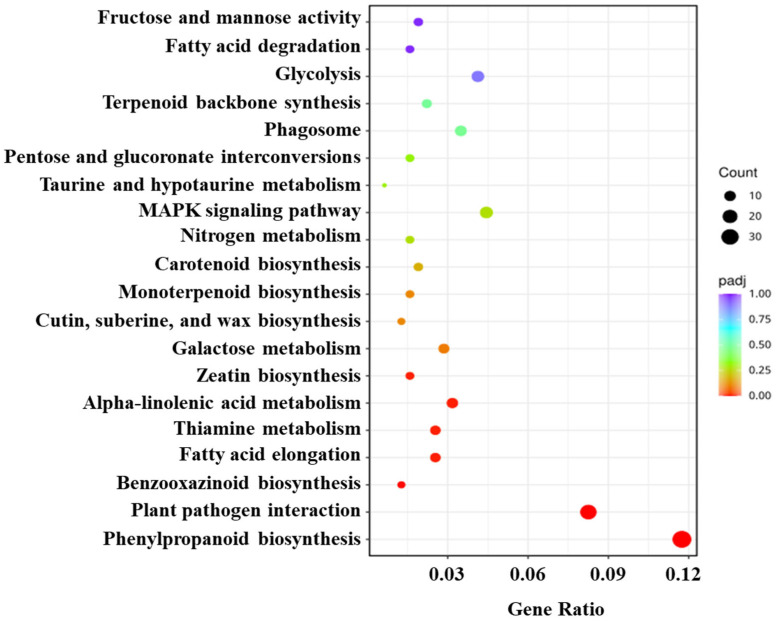
KEGG pathway classification and functional enrichment of differentially expressed genes in sugarcane response to SrMV infection and SA treatment.

**Figure 4 plants-13-00234-f004:**
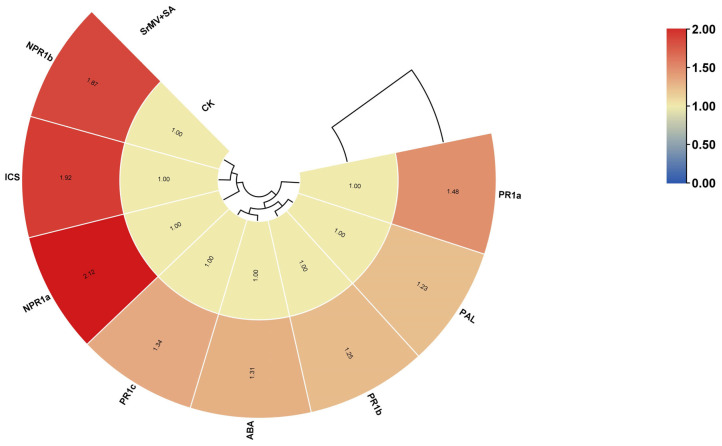
Transcript expression of selected candidate genes from RNA-seq dataset.

**Figure 5 plants-13-00234-f005:**
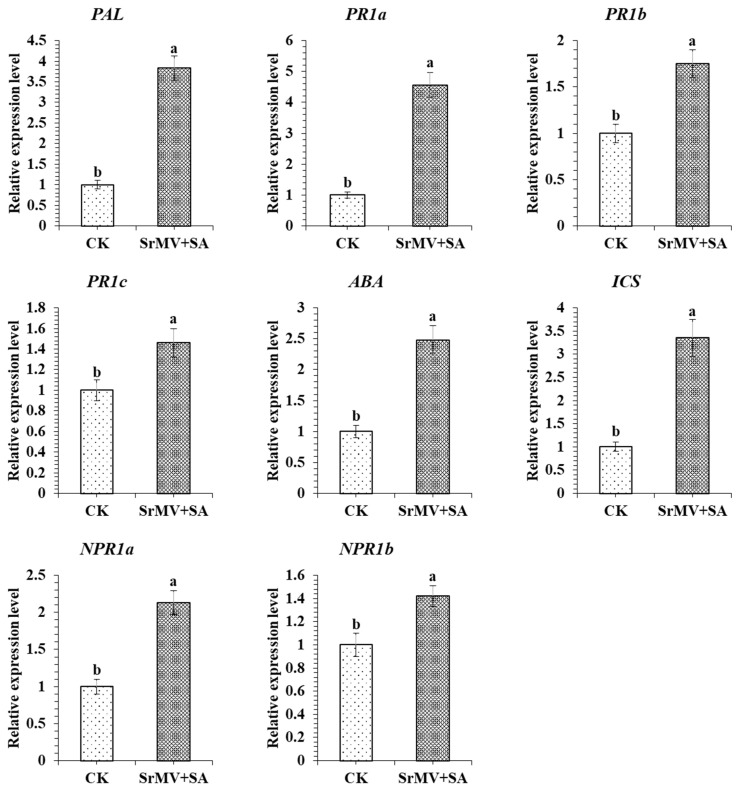
Transcript profiling of candidate genes by qRT-PCR assay. Values for relative expression levels are means ± standard errors. Different lowercase letters above the bars are significantly different at a 5% probability level.

**Table 1 plants-13-00234-t001:** Summary of transcriptome sequencing and assembly of sugarcane in response to SrMV infection and SA application.

Sample	Raw Reads	Clean Reads	Q20 (%)	Q30 (%)	GC (%)	Mapping Rate (%)
CK-1	44,914,726	43,314,264	97.5	93.34	51.76	85.01
CK-2	40,068,588	37,546,314	97.5	93.28	49.83	87.59
CK-3	41,815,628	40,116,578	97.36	93.24	56.06	86.90
CK-4	46,929,116	45,112,170	97.48	93.22	50.38	81.90
CK-5	43,052,022	41,398,032	97.6	93.42	48.49	87.37
CK-6	44,478,236	42,657,658	97.47	93.26	51	85.09
SA-1	41,941,044	40,182,148	97.43	93.31	53.92	81.29
SA-2	44,570,072	42,837,078	97.36	93.14	54.66	83.18
SA-3	45,719,490	44,294,824	97.39	93.19	53.73	78.52
SA-4	44,841,752	43,183,362	97.28	93.01	55.88	89.37
SA-5	40,056,842	38,165,778	97.38	93.25	55.25	84.26
SA-6	47,349,582	45,324,754	97.41	93.34	56.06	87.88

Note: Q20%, and Q30%: respectively refer to the percentage of bases with sequencing quality above 99% and 99.9% of the total bases; GC%: is the percentage of the sum of G and C bases corresponding to the quality control data to the total bases. SA solution with a concentration of 1 mM was prepared and applied to sugarcane seedlings following SrMV inoculation with 6 biological replicates (SA-1 to SA-6).

## Data Availability

Data are contained within the article and Appendix A.

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
