# Peer review of "Transcriptomic Analysis Reveals Candidate Genes in Response to Sorghum Mosaic Virus and Salicylic Acid in Sugarcane"

_plants, 2024, doi:10.3390/plants13020234_

Round 1
Reviewer 1 Report
Comments and Suggestions for Authors
The MS entitled "Transcriptomic Analysis Reveals Candidate Genes in Response to Sorghum Mosaic Virus and Salicylic Acid in Sugarcane" utilizes the transcriptomic approach to identify candidate genes involved in SrMV resistance after SA treatment. The MS structure is well-organized and the experimental design is well thought. I have included my comments in the PDF file. Authors are suggested to read the comments and make necessary changes in the MS.

Author Response
The MS entitled "Transcriptomic Analysis Reveals Candidate Genes in Response to Sorghum Mosaic Virus and Salicylic Acid in Sugarcane" utilizes the transcriptomic approach to identify candidate genes involved in SrMV resistance after SA treatment. The MS structure is well-organized and the experimental design is well thought. I have included my comments in the PDF file. Authors are suggested to read the comments and make necessary changes in the MS.
Response: Esteemed reviewer, thank you for your valuable suggestions and comments. We have edited the manuscript based on your suggestions/comments. We believe that the corrections will meet with the approval.
Reviewer 2 Report
Comments and Suggestions for Authors
This manuscript (MS) describes the effects of sorghum music virus and SA treatment and transcriptome analysis in sugarcane. While similar studies have been carried out in several other plant species to date, this appears to be the first time it has been done in sugarcane. However, several statements in the discussion were based on previous studies, and these should be revised (details in comments below) with the current study and synthesis stories. Overall, the current MS objectives are simple, and with limited analyses, further experiment or analysis is required to mold a story to gain the reader's interest.
Major comments:
1. It is a well-known phenomenon that the application of SA triggers an immune response in plants. The current study is not meaningful, a readers find the novel information. The MS needs to clarify from the reader's perspective.
2. Gene enrichment by the KEGG pathway, was not given any novel information apart from the candidate gene reported here as regular WRKY, PR1, and NPR1.
3. Experiment to analyze the list of differentially expressed transcription factors (TFs) involved, mine the cis-elements in their promoter, and correlate those cis-elements in candidate gene promoter sequence from the group of Phenylpropanoid biosynthesis and plant pathogens interaction. They will give stories about infection of the virus along with SA treatment.
4. The discussion section needs to be significantly revised, and presently, with a general discussion of already existing information.
5. Autor should perform MapMan with the list of upregulated and downregulated Transcription factors and candidate genes, which will give an idea of any novel gene encoding transcript and transcription factors that modulated differential response to virus infections and SA. This could be done BLAST reference genes using Arabidopsis or rice TFs and gene information.
6. The author should be given a list of upregulated and downregulated information with read counts, FPKM, and fold changes as supplemental information so that this will be helpful for those who are working in sugarcane for further studies.
Comments on the Quality of English Languageminor editing required
Author Response
This manuscript (MS) describes the effects of sorghum music virus and SA treatment and transcriptome analysis in sugarcane. While similar studies have been carried out in several other plant species to date, this appears to be the first time it has been done in sugarcane. However, several statements in the discussion were based on previous studies, and these should be revised (details in comments below) with the current study and synthesis stories. Overall, the current MS objectives are simple, and with limited analyses, further experiment or analysis is required to mold a story to gain the reader's interest.
Major comments:
- It is a well-known phenomenon that the application of SA triggers an immune response in plants. The current study is not meaningful, a readers find the novel information. The MS needs to clarify from the reader's perspective.
Response: Esteemed reviewer, thank you for your valuable suggestion. To our knowledge, no report relates to an integrated transcriptome analysis of SrMV infection and SA application in sugarcane. Therefore, current study first time describes the coupling effect of SA application and SrMV infection. Based on your suggestion, we have clarified the objectives of the current study in a better way to improve the quality in the reader’s perspective.
- Gene enrichment by the KEGG pathway, was not given any novel information apart from the candidate gene reported here as regular WRKY, PR1, and NPR1.
Response: Dear reviewer, to explore the biological pathways triggered by SrMV infection and SA application, DEGs were annotated by KEGG enrichment analysis. KEGG enrichment showed that most of the DEGs are mainly concentrated in metabolic pathways and plant-pathogen interaction, suggesting that those pathway may be involved in SA-induced disease resistance of sugarcane. From those interesting pathways in defense mechanism we have selected some interesting candidate genes for checking their transcript profiles.
- Experiment to analyze the list of differentially expressed transcription factors (TFs) involved, mine the cis-elements in their promoter, and correlate those cis-elements in candidate gene promoter sequence from the group of Phenylpropanoid biosynthesis and plant pathogens interaction. They will give stories about infection of the virus along with SA treatment.
Response: Thank you for your valuable suggestion. You are right. Based on this transcriptome results, we further identified two complete gene families (WRKY and PR1). We have done several important analysis such as genes structure, cis-elements, collinearity and gene duplication events. Those gene families are separate two papers and the addition of analysis in this ms will decrease the worth of those gene families’ identification and characterization. For your reference, we have given the results by the end of this response report.
- The discussion section needs to be significantly revised, and presently, with a general discussion of already existing information.
Response: We have significantly revised the discussion section based on your suggestion.
- The author should be given a list of upregulated and downregulated information with read counts, FPKM, and fold changes as supplemental information so that this will be helpful for those who are working in sugarcane for further studies.
Response: We have added a figure based on your suggestion in main text.
Reviewer 3 Report
Comments and Suggestions for Authors
Dear Authors,
The subject of the study is interesting and topical, with scientific and practical importance.
The introduction is presented correctly, in accordance with the subject. Numerous scientific articles, in concordance to the topic of the study, were consulted.
Methodology of the study was clearly presented, and appropriate to the proposed objectives.
The obtained results are important and have been analyzed and interpreted correctly, in accordance with the current methodology.
The discussions are appropriate, in the context of the results, and was conducted compared to other studies in the field.
The scientific literature, to which the reporting was made, is recent and representative in the field.
Some suggestions and corrections were made in the article.
The following aspects are brought to the attention of the authors.
1.
Please check the writing of the units of measure to the values in table 1, in accordance with the Instructions for Authors and Microsoft Word template, Plants journal
2.
References
According to Instructions for Authors, and Microsoft Word template, Plants journal
“Author 1, A.B.; Author 2, C.D. Title of the article. Abbreviated Journal Name Year, Volume, page range.”
e.g. Page 11
a)
“Kummu, M.; Heino, M.; Taka, M.; Varis, O.; Viviroli, D. Climate change risks pushing one-third of global food production outside the safe climatic space. One Earth 2021, 4(5), 720-729. Doi. https://doi.org/10.1016/j.oneear.2021.04.017”
instead of
“Kummu, M., et al., Climate change risks pushing one-third of global food production outside the safe climatic space. One Earth, 2021. 4(5): p. 720-729.”
b)
“Javed, T.; Gao, S.-J. WRKY transcription factors in plant defense. Trends Genet. 2023, 39(10), 787-801. Doi. https://doi.org/10.1016/j.tig.2023.07.001”
Instead of
“Javed, T. and S. Gao, WRKY transcription factors in plant defense. Trends in Genetics, 2023. 39(10): p. 787-801.”
Abbreviated Journal Name
Year
Volume
It is recommended to review the entire References chapter, and to correct it, if necessary.

Author Response
Dear Authors,
The subject of the study is interesting and topical, with scientific and practical importance.
The introduction is presented correctly, in accordance with the subject. Numerous scientific articles, in concordance to the topic of the study, were consulted.
Methodology of the study was clearly presented, and appropriate to the proposed objectives.
The obtained results are important and have been analyzed and interpreted correctly, in accordance with the current methodology.
The discussions are appropriate, in the context of the results, and was conducted compared to other studies in the field.
The scientific literature, to which the reporting was made, is recent and representative in the field.
Some suggestions and corrections were made in the article.
The following aspects are brought to the attention of the authors.
1.
Please check the writing of the units of measure to the values in table 1, in accordance with the Instructions for Authors and Microsoft Word template, Plants journal
Response: Dear reviewer, thank you for valuable suggestion. We have followed the instruction given by the journal as per your suggestion.
2.
References
According to Instructions for Authors, and Microsoft Word template, Plants journal
“Author 1, A.B.; Author 2, C.D. Title of the article. Abbreviated Journal Name Year, Volume, page range.”
e.g. Page 11
a)
“Kummu, M.; Heino, M.; Taka, M.; Varis, O.; Viviroli, D. Climate change risks pushing one-third of global food production outside the safe climatic space. One Earth 2021, 4(5), 720-729. Doi. https://doi.org/10.1016/j.oneear.2021.04.017”
instead of
“Kummu, M., et al., Climate change risks pushing one-third of global food production outside the safe climatic space. One Earth, 2021. 4(5): p. 720-729.”
b)
“Javed, T.; Gao, S.-J. WRKY transcription factors in plant defense. Trends Genet. 2023, 39(10), 787-801. Doi. https://doi.org/10.1016/j.tig.2023.07.001”
Instead of
“Javed, T. and S. Gao, WRKY transcription factors in plant defense. Trends in Genetics, 2023. 39(10): p. 787-801.”
Abbreviated Journal Name
Year
Volume
It is recommended to review the entire References chapter, and to correct it, if necessary.
Response: Dear reviewer. We have edited all the references as per journal style. Thank you for your valuable suggestion
Round 2
Reviewer 2 Report
Comments and Suggestions for Authors
The revised manuscript is well structured and written, and the conclusions are supported by the analysis of the data presented therefore, the paper can be accepted for publication after considering my comments below:
The author should be given the spreadsheet containing a list of upregulated and downregulated information with read counts, FPKM, and fold changes as supplemental information so that this will be helpful for those who are working in sugarcane for further studies. The given figures are not sufficient for readers, therefore providing the spreadsheet with the necessary information mentioned above.
Author Response
The revised manuscript is well structured and written, and the conclusions are supported by the analysis of the data presented therefore, the paper can be accepted for publication after considering my comments below:
The author should be given the spreadsheet containing a list of upregulated and downregulated information with read counts, FPKM, and fold changes as supplemental information so that this will be helpful for those who are working in sugarcane for further studies. The given figures are not sufficient for readers, therefore providing the spreadsheet with the necessary information mentioned above.
Response: Esteemed reviewer, thank you for your valuable suggestion. We have provided the complete information as supplementary file which is available online (https://figshare.com/search?q=10.6084%2Fm9.figshare.24816597) based on you query.